# A New, Efficient Conversion Technology to Transform Ambient CO₂ to Valuable, Carbon-Based Fuel via Molten Salt Electrochemistry

**Deqiang Ji** 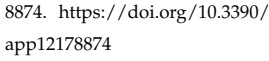**, Qingxin Jia, Chuanli Zhu, Wei Dong, Hongjun Wu * and Guanzhong Wang ***

Key Laboratory of Energy Transformation and Utilization Technology, College of Chemistry and Chemical Engineering, Northeast Petroleum University, Daqing 163318, China
* Correspondence: hjwu@nepu.edu.cn (H.W.); wangguanzhong0718@gmail.com (G.W.)

**Abstract:** Climate warming and environmental problems caused by the excessive consumption of fossil energy and massive CO₂ emissions have seriously damaged the human living environment. This paper develops a new green, efficient, and environmentally friendly CO₂ capture and conversion method, which is a crucial way to alleviate the greenhouse effect. In this study, alkali metal carbonates (and the corresponding hydroxides) are fused and blended to construct a liquid molten salt electrolyte system with excellent performance, which is applied to synthesize carbon materials or carbon-based fuel gas. By regulating the electrolyte composition and electrolysis parameters, carbon-based fuels with different micro-morphologies and compositions can be prepared in a controllable manner. In pure Li₂CO₃ electrolyte, carbon nanotubes (CNTs) with a high value are synthesized at 750 °C with, initially, 10 mA/cm² and, finally, with 100 mA/cm². Carbon spheres are obtained in Li-Ca-Ba at 750 °C with 200 mA/cm², while honeycomb carbon is generated in the electrolyte of Li-Na-K at 450 °C with 450 mA/cm². Syngas (33.6%) or CH₄-rich fuel gas (40.1%) can also be obtained by adding LiOH into the electrolyte under 500 °C at 2.0 V and 3.2 V, respectively. This paper provides a new way of utilizing CO₂ resources and a new sustainable green development.

**Keywords:** electrolyte of carbonate and hydroxide; electrochemical reduction of CO₂; co-electrolysis of CO₂ and H₂O; carbon-based materials; syngas and CH₄-rich fuel gas

## 1. Introduction

The excessive reliance on burning carbon-heavy fossil fuels has undoubtedly given rise to numerous environmental crises. The predominant one is the rapidly rising concentration of CO₂ in the atmosphere, which is considered to be a dominant factor contributing to climate change, especially the greenhouse effect and global warming [1–3]. There have been multiple energy storage [4–7] and physical storage [8,9] approaches to keeping the post-combustion CO₂ emissions within limits, but it appears to have had little effect. By contrast, consumption of the greenhouse gas CO₂ to manufacture high added-value sustainable energy by chemical synthesis, including carbon nanomaterials [10], methane [11,12], carbon monoxide [13], and other carbonaceous compounds [14,15], can be an effective measure to alleviate global warming.

For the chemical activation of CO₂, the first step is the stripping of an oxygen atom from CO₂ to generate syngas (Equation (1)), which can be converted to methanol (Equation (2)), a liquid organic substance that can be directly utilized or secondarily converted to other valuable chemicals and fuels [16].

$$CO_2 + 2e^- = CO + O^{2-} \tag{1}$$

$$CO + 2H_2 = CH_3OH \tag{2}$$

This technique is conducive to transforming $CO_2$ to carbonic products, but it is subject to the high activity and selectivity catalyst [17]. Furthermore, the high hydrogen consumption also restricts the industrialized process of $CO_2$ hydrogenation. Similarly, a variety of hydrocarbons can be synthesized via the Fischer–Tropsch reaction (Equation (3)), but the high price of $H_2$ and CO has become a huge obstacle to further development. [18,19].

$$nCO + (2n + 1)H_2 = C_nH_{(2n+2)} + nH_2O \tag{3}$$

We demonstrate a new design of the direct synthesis of valuable, carbon-based materials (carbon nanomaterials, methane, and syngas) from ambient $CO_2$ using eutectic molten salt electrolytes in an alumina reactor. In our study, the broad-based feedstock $CO_2$ was transformed into high-value, objective products without a noble metal electrocatalyst at a rate of 100 times faster than that observed in photoelectrochemical systems or fuel cell systems. More importantly, the Fischer–Tropsch reaction was no longer essential to the generation of methane. Syngas or methane can be synthesized via a one-step co-electrolysis of reducible C (from the $CO_3^{2-}$ in the electrolyte or dissolved $CO_2$) and monovalent H (from the OH⁻ in the electrolyte or captured $H_2O$) in eutectic molten salts. A high-value carbon nanotube content of ~80% was observed in the electrolysis of pure $Li_2CO_3$ at 750 °C with increasing current density, starting from 10 mA/cm² to 100 mA/cm². In the co-electrolysis of the $CO_2$ and $H_2O$ system, the $CH_4$ content can reach 40.1% at 550 °C under 2.0 V, while the CO content can rise up to 33.6% at 550 °C under 3.5 V in the electrolyte of $Li_{1.43}Na_{0.26}K_{0.21}CO_3$ + LiOH (90%: 10%, wt%). This new study suggests an innovative, sustainable path for the capture and conversion of $CO_2$ driven by electrical energy or other renewable sources.

## 2. Materials and Methods

**Chemicals.** The diverse weight ratio of carbonates (and hydroxides), allowing the continuous absorption and electrochemical transformation of $CO_2$ (and $H_2O$) into valuable carbon-based fuels, is contained in a long-life corundum crucible. The electrolyte of lithium carbonate ($Li_2CO_3$, 99.99%, battery grade), sodium carbonate ($Na_2CO_3$, 99.5%), potassium carbonate ($K_2CO_3$, 99%), calcium carbonate ($CaCO_3$, 99%), barium carbonate ($BaCO_3$, 99%), and lithium hydroxide (LiOH, 99.9%) were provided by Beijing InnoChem and Macklin. The electrode system of spiral Fe wire cathode (active area, 10 cm²) and spiral Ni wire anode (active area, 10 cm²) was purchased from Shanghai Makclin Biochemical Co., Ltd. and was polished to avoid the influence of the oxide layer on the metal surface.

**Method.** The corundum crucible, containing eutectic molten salts, was placed inside a ceramic heater powered by electricity. To ensure that the moisture of the electrolyte was removed completely, the mixed molten salts were kept at operating temperature (50 °C above the melting point, at least) for 3 h. The electrolysis of the $CO_2$ (and $H_2O$) from $CO_3^{2-}$ (and H⁺) in the eutectic molten salts occurred at the operating current density for the desired duration. The cathodic solid products in the system of pure carbonates were collected through a series of purification treatments: acid leaching → ultrasound process → filtration → water bath (deionized water) → drying. The gaseous products released from the sealed reactor through a gas-guide tube were collected in the airbag for composition analysis. Even more importantly, a well-sealed stainless-steel reactor was used to ensure the accuracy of the product composition.

**Characterization.** The micro-structure of the solid carbon on the cathode was observed by a Zeiss SIGMA SEM (scanning electron microscope) with a 15 KV working voltage. With the help of the the EDS (energy dispersive spectroscope) component of the SEM, the element composition of the cathodic solid was detected. Thus, the gaseous products, collected in the air bag, were analysed by gas chromatography (Agilent 7890B). The $H_2$, $O_2$, and $CH_4$ were detected by a thermal conductivity detector (TCD).

## 3. Results and Discussion

### 3.1. Reaction Mechanism of Single $CO_2$ and $CO_2$ and $H_2O$

It is well known that the traditional $CO_2$ activation and conversion methods are limited by the high stability and low conversion of $CO_2$. Under standard conditions, only $1.3 \times 10^{-5}$ moles of carbon per liter of air are available for the reaction. As high as 28.5 moles o reducible $CO_3^{2-}$, nearly seven orders of magnitude more than the air, is in one liter of molten $Li_2CO_3$ at 723 °C. The results show that the reducible carbon concentration of carbonate anions is much higher than that of airborne or dissolved carbon dioxide, which improves the mass transfer efficiency.

The movement, reduction, and absorption of carbon dioxide are carried out by the electrolysis of tetravalent carbon to zero-valent carbon or other carbon-based fuels on the cathode's surface [20,21]. There is a general belief that the electrochemical reaction of $CO_2$ in molten salts occurs in two different reaction mechanisms: the direct path and the indirect path [20]. As shown in the below equations, more authors are inclined to recognize that elemental carbon is generated indirectly in molten salt by $CO_2$ through $CO_3^{2-}$. Alkali or alkaline earth metals, such as Li, Na, and K, are represented by M. The most apparent difference between the reduction mechanisms is the first step, the transformation from $CO_2$ to charged $CO_3^{2-}$ (Equation (4)). Then, the newly formed $CO_3^{2-}$ from the $CO_2$ and the original ionized $CO_3^{2-}$ from the electrolytes can be reduced to elemental C or bivalent CO and oxide ions via Equation (5) or (6). There are two different reaction trends for $O^{2-}$ that is formed simultaneously with the reduction products. One part of the $O^{2-}$ moves to the anode, where it loses two electrons and converts into $O_2$ (Equation (7)) [22]. The rest of the $O^{2-}$ further acts as an absorbent for $CO_2$, allowing the regeneration of molten carbonate electrolytes (Equation (7)) [23]. The $H_2O$ is also transformed into $H_2$ and $O_2$ through the process that resembles that of $CO_2$ [24,25].

First Step: absorption of $CO_2$ in molten salts

$$M_2O + CO_2 = M_2CO_3 \tag{4}$$

Second Step: reduction of $CO_3^{2-}$ to CO or C

$$CO_3^{2-} + 2e^- = CO + 2O^{2-} \tag{5}$$

$$CO_3^{2-} + 4e^- = C + 3O^{2-} \tag{6}$$

Oxidation of $O^{2-}$

$$O^{2-} = O_2 + 4e^- \tag{7}$$

Regeneration of $CO_3^{2-}$:

$$CO_2 + O^{2-} = CO_3^{2-} \tag{8}$$

In this study, the electrochemical reduction of a single $CO_2$ or $CO_2$ and $H_2O$ is driven by applying an external electric field to the sealed reaction electrolyzer. The electrical energy consumed in this process can be replaced by sustainable energy sources, including solar, wind, and tidal power. As shown in Scheme 1, the environmental contaminant $CO_2$ is transformed into solid carbon or co-electrolyzed with convenient $H_2O$ to generate syngas or $CH_4$ by high efficiency and selectivity on the cathode's surface. As mentioned above, the electrolyte can be regenerated through the rapid absorption capacity of the intermediates (metal oxides) for $CO_2$ and $H_2O$, completing the construction of the circulatory system.

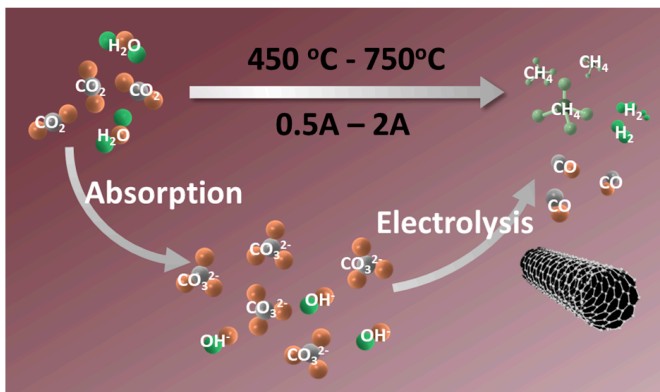

**Scheme 1.** The route for the transformation of $CO_2$ and $H_2O$ into carbon-based fuels.

### 3.2. The Cathodic Production of Carbon

It is well known that the melting points of pure $Li_2CO_3$, $Na_2CO_3$, and $K_2CO_3$ are relatively high, up to 723 °C, 851 °C, and 891 °C, respectively [26], signifying that over 950 °C is required for electrolysis (at least 50 °C higher than the melting point). As stated in our previous works, the pure carbonates will decompose to metallic oxide and carbon dioxide at excessively high temperatures [24,25]. The high operating temperature inevitably leads to unwanted thermal decomposition and the waste of raw materials [25,27,28]. To reduce the working temperature and enable less energy consumption, we employed the ternary eutectic mixture of $Li_2CO_3$, $Na_2CO_3$, and $K_2CO_3$ as the electrolyte to transform tetravalent carbon to solid carbon at a lower temperature, such as $Li_{1.43}Na_{0.26}K_{0.21}CO_3$ (61:22:17, mass ratio), which melts at 390 °C, and $Li_{0.85}Na_{0.61}K_{0.54}CO_3$ (31:32:37, mass ratio), which melts at 393 °C.

The composition of molten salts has been reported to affect the carbon electrodeposition from $CO_3^{2-}$ [29]. In our previous works, no amorphous carbon was obtained in pure $Na_2CO_3$, $K_2CO_3$, or a blend of both [23], clearly suggesting that the role of individual carbonates in the $CO_2$ electrochemical process is quite different. The thermodynamic calculation in Table 1 can explain why the reduction of carbon from $Li_2CO_3$ is more favorable than that from $Na_2CO_3$ or $K_2CO_3$ under certain conditions.

**Table 1.** Standard potentials for the conversion of alkali metal carbonate to pure alkali metal, carbon, and carbon dioxide ($Li_2CO_3$, $Na_2CO_3$, or $K_2CO_3$ at different temperatures).

| T (K) | Carbonate | $E^\circ_M$ (V) | $E^\circ_C$ (V) | $E^\circ_{CO}$ (V) |
|---|---|---|---|---|
| | $Li_2CO_3$ | 3.218 | 1.901 | 2.318 |
| 700 | $Na_2CO_3$ | 2.791 | 2.706 | 3.392 |
| | $K_2CO_3$ | 2.853 | 3.249 | 4.116 |
| | $Li_2CO_3$ | 3.145 | 1.845 | 2.221 |
| 750 | $Na_2CO_3$ | 2.719 | 2.655 | 3.301 |
| | $K_2CO_3$ | 2.778 | 3.191 | 4.014 |
| | $Li_2CO_3$ | 3.071 | 1.789 | 2.124 |
| 800 | $Na_2CO_3$ | 2.649 | 2.603 | 3.209 |
| | $K_2CO_3$ | 2.703 | 3.132 | 3.913 |
| | $Li_2CO_3$ | 2.998 | 1.735 | 2.029 |
| 850 | $Na_2CO_3$ | 2.579 | 2.552 | 3.118 |
| | $K_2CO_3$ | 2.631 | 3.074 | 3.813 |
| | $Li_2CO_3$ | 2.926 | 1.681 | 1.933 |
| 900 | $Na_2CO_3$ | 2.508 | 2.502 | 3.027 |
| | $K_2CO_3$ | 2.557 | 3.015 | 3.712 |

The relevant thermodynamic data deposited to metal, carbon, or carbon dioxide from a single $Li_2CO_3$, $Na_2CO_3$, or $K_2CO_3$ at different temperatures is given in Table 1. After analyzing the above calculation results, we found that the potential required to deposit carbon ($E°_C$) or CO ($E°_{CO}$) is lower than that for Li metal deposition ($E°_M$) in pure molten $Li_2CO_3$, revealing a stronger tendency for the carbon electrodeposition. However, the $E°_C$ of $Na_2CO_3$ and $K_2CO_3$ is comparable or more positive than the $E°_M$, indicating that metal deposition is more preferred than carbon in $Na_2CO_3$ or $K_2CO_3$. The calculated results shown in the above table indicate that the generation of lithium metal from $Li_2CO_3$ needs a higher electrolysis potential, and the electrodeposition of $Li_2CO_3$ into carbon is much more favorable in comparison to $Na_2CO_3$ and $K_2CO_3$. Therefore, $Li_2CO_3$ is the most unfavorable ingredient for metal precipitation and becomes the main body of eutectic molten salts.

It has been reported that $CO_2$ can be transformed into solid carbon in molten salts under $N_2$, $CO_2$, or Ar [30,31]. Under the appropriate conditions, regardless of the gaseous environment ($N_2$, $CO_2$, or Ar), there is a transformation of $CO_3^{2-}$ ions into nonvalent carbon in molten lithium-rich carbonate. The electrolysis system shows a ~95% current efficiency in $Li_2CO_3$-$Na_2CO_3$-$K_2CO_3$ under Ar, while it is only 67% under $CO_2$ due to the thermal reductive reaction of carbon with $CO_2$ to CO [29]. Although the increasing $CO_2$ partial pressure can enhance the carbon electrodeposition rate, it will lead to a huge loss of energy. Argon exhibits an excellent performance in preventing or inhibiting the corrosion of electrodes, and it can also separate the gaseous oxidation products. With that in mind, $Li_{1.43}Na_{0.26}K_{0.21}CO_3$ is the optimal electrolyte to make the whole process of transforming $CO_2$ into carbon products in the atmosphere of argon work effectively.

Figure 1 shows a CV curve (cyclic voltammetry) of an Ni anode and Fe cathode in t $Li_{1.43}Na_{0.26}K_{0.21}CO_3$ at 550 °C. No reduction peak is observed between 0 V and 1 V at any scanning rate in Figure 1a, suggesting that no reduction product is obtained in the low-potential range. The first reduction peak, R1, appears at 1.5 V, indicating carbon deposition. With the continually rising potential, the following reduction peak, R2 at 2.3 V, represents the generation of CO from the molten carbonates. It can be seen from the reduction peak R3 that the maximal electrolysis potential is 2.8 V. Otherwise, the alkaline metals (Na and K) will be deposited on the surface of the cathode and result in wasteful power consumption. The oxidation peaks C1 and C2 clearly suggest the electrolysis dissolution of the deposited alkaline metal and carbon, which can be verified by the CVs of successive scans (Figure 1b) [32]. As shown in Figure 1b, the reduction current of R1 and the anodic current of C1 increase with the scanning cycles, which can be explained by continuous carbon deposition, resulting in the growing surface of the working electrodes [33].

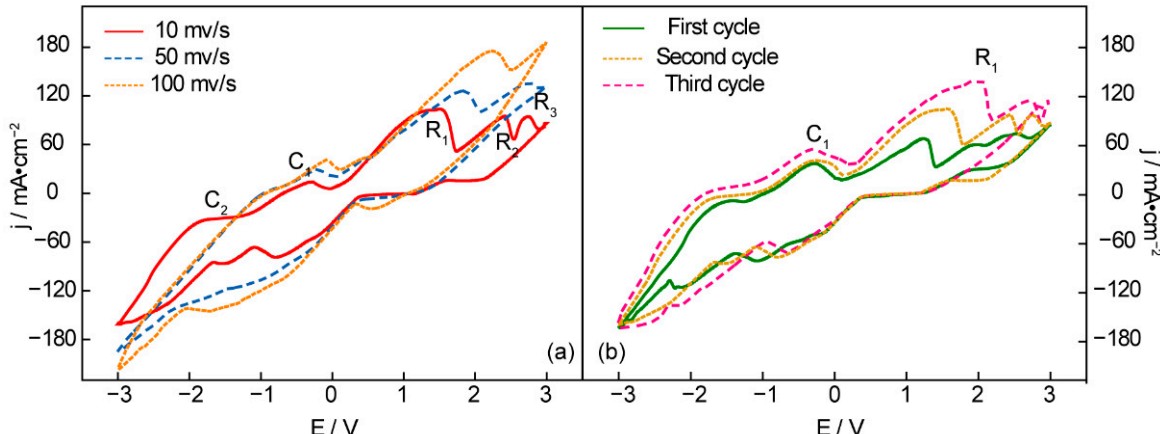

**Figure 1.** Cyclic voltammetry recorded from an Ni anode and Fe cathode in $Li_{1.43}Na_{0.26}K_{0.21}CO_3$ at 550 °C under consecutive cycling (**a**) in different scanning rates and (**b**) different scanning cycles at 10 mv/s.

The electrolysis experiment was conducted to further investigate the carbon materials deposited on the Fe cathode in the molten $Li_{1.43}Na_{0.26}K_{0.21}CO_3$ with the Ni anode at 550 °C, and the result is shown in Figure 2. The cell–current curve in Figure 2 was recorded during the electrolysis reaction at 2.0 V with an electrochemical workstation, and the electrode photos obtained at the different reaction intervals during the electrolysis process at 2.0 V are shown on the timeline. As can be seen in the digital photo of the cathodes, a layer of carbon particles is observed on the surface of the Fe cathode, revealing an increasing tendency of carbon products as the reaction time increases. Obviously, cell-current values also increase as time goes on, which is consistent with the anticipated increase in the electrode's surface area. In the electrolyte of $Li_{1.43}Na_{0.26}K_{0.21}CO_3$ at certain conditions, as above-mentioned, the synthesis carbon comprises five elements (mole ratio): C (94%), O (5%), Na (0.3%), Au (0.3%), and Cl (0.3%). In particular, it needs to be explained that the peak of the Au element comes from the gold spraying operation before the SEM characterization. There is a predominant mole fraction (94%) of C to that of O in the solid black product, suggesting that the cathodic product is likely solid carbon, with a few oxygen-containing functional groups [34]. The small number of Na or Cl detected in the carbon products may come from the molten salts that have not been thoroughly cleaned.

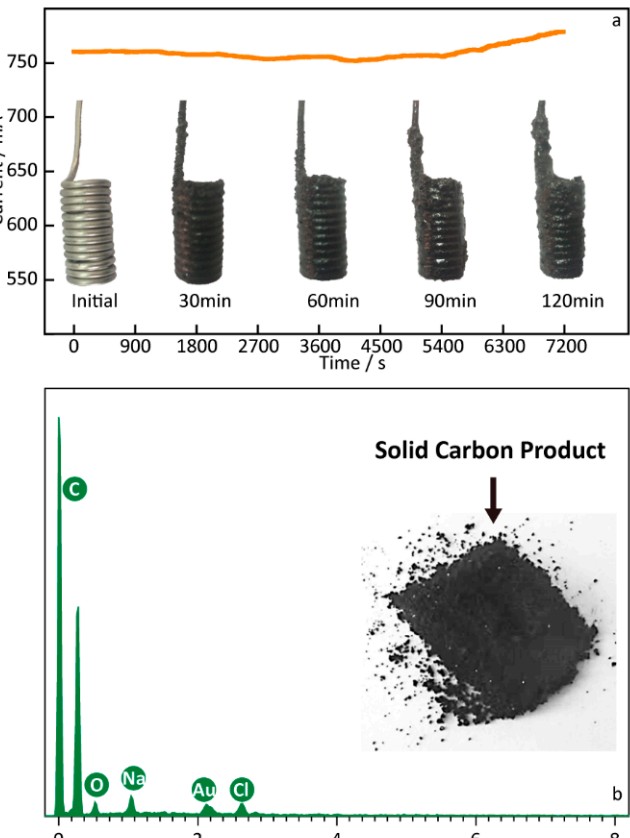

**Figure 2.** (**a**) Cell–current curve and digital photos of the Fe cathode after different electrolysis times at 550 °C. (**b**) EDS analysis of the solid carbon obtained from the $Li_{1.43}Na_{0.26}K_{0.21}CO_3$ electrolysis at 2.5 V.

As in previous studies, the minimum theoretical electrolytic potential for carbon deposition is 1.65 V at 550 °C in a pure $Li_2CO_3$ electrolyte. The working voltage was determined to be 2.5 V to ensure that the elemental carbon can be reduced on the cathode's surface. Considering the electrical resistance of molten salts and the corrosion resistance of electrolytic cells, 500 °C to 750 °C was selected to investigate the microstructure characteristics of carbon, and the SEM photographs are shown below in Figure 3a–d.

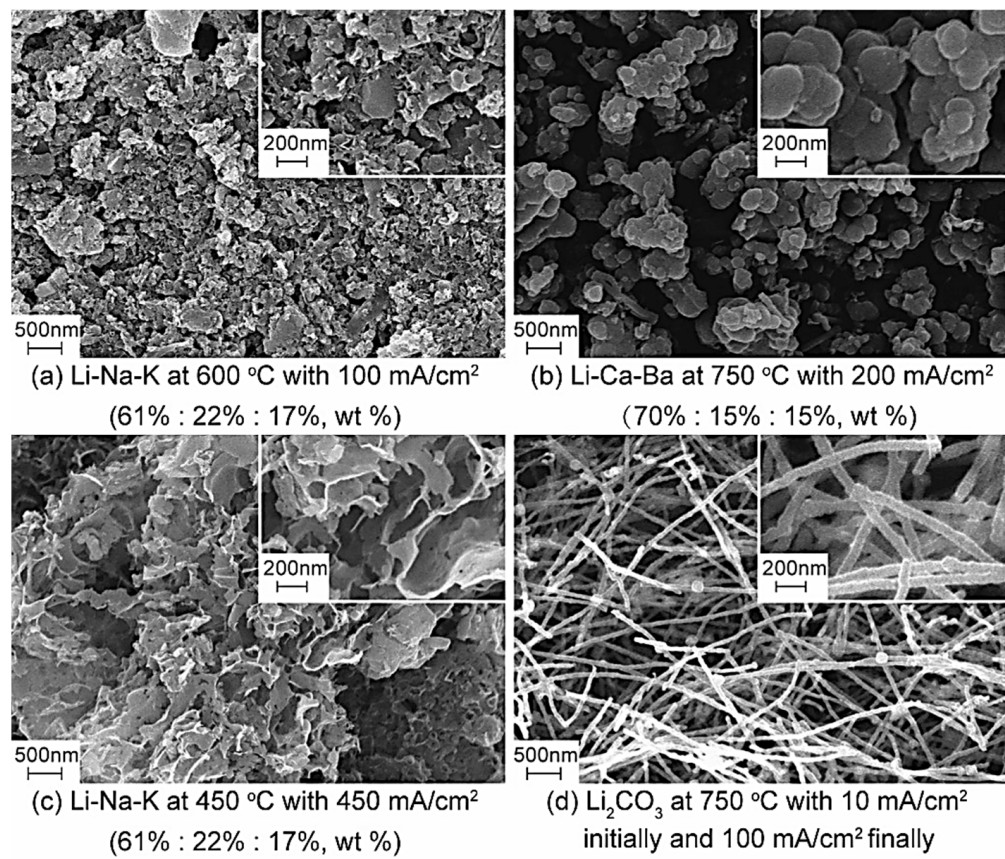

**Figure 3.** SEM of the carbon materials under different conditions.

The carbon nanomaterial is a class of carbon material having at least one size less than 100 nm on the dispersed phase scale. The price of carbon nanomaterials is $10,000 to $400,000 per ton, approximately 10 to 400 times more than regular activated carbon. More importantly, the economic and application value of carbon nanomaterials is much higher than that of carbon dioxide material. It is evident from Figure 3a that amorphous carbon materials with unique microstructures can be obtained in an $Li_2CO_3$-$Na_2CO_3$-$K_2CO_3$ electrolyte at 600 °C with 100 mA/cm$^2$. As can be seen intuitively in Figure 3b, a large number of spherical carbon particles with different diameters (micron scale) obtained at different cell voltages are irregularly stacked together, showing similar microstructures and appearing like aggregates of carbon nanoparticles. The main reason for producing carbon spheres is the interfacial effect of Cao and BaO, which is formed during electrolysis. The diameter of a carbon sphere product is approximately 200–300 nm. Honeycomb carbon is another product with a unique microscopic morphology, which is generated via electrolysis in alkali carbonates at 450 °C with 450 mA/cm$^2$. By regulating the electrolysis temperature and current density, carbon materials with high surface area, up to 200 m$^2$/g, can be separated from the cathode's surface. More importantly, carbon nanotubes (CNTs) are observed in the SEMs of products obtained in pure $Li_2CO_3$ at 750 °C with 10 mA/cm$^2$, initially, and 100 mA/cm$^2$, finally. The purity of CNTs with a diameter of 50–80 nm is up to 80%, and their current efficiency can reach more than 85%. It has been demonstrated that molten salt electrochemical technology has the potential to electrochemically reduce $CO_2$ to high-value, carbon-based materials.

An X-ray diffractometer (XRD) and Raman spectroscopy were employed to further characterize and analyze carbon nanotubes, which is shown in Figure 4. In the left of Figure 4, a significantly enhanced and narrow characteristic peak (red star) of carbon nanotubes is observed at 26° (002), indicating that the graphite hexagonal crystals in the CNTs are relatively regular and the degree of crystallization is high. The carbon atoms in the wall of the carbon nanotubes are mainly sp$^2$ hybridized, with only a small amount

being sp³ hybridized. Because of their unique crystalline form, CNTs have a high modulus and strength along the axis, which can be used to enhance the mechanical properties of composites. In the right of Figure 4, the Raman peaks at 1350 cm⁻¹ and 1580 cm⁻¹ correspond to the characteristic disorder-induced mode (D-band) and the high-frequency $E_{2g}$ first-order mode (G-band), respectively. The peak G intensity of the carbon nanotubes synthesized in pure $Li_2CO_3$ is significantly higher than that of peak D ($I_D/I_G < 1$), which means that the degree of graphitization is high and the arrangement of graphite hexagonal crystal formed by the carbon atoms is regular and orderly. The $I_D/I_G$ value is between 0.7 and 0.8, which is completely consistent with commercial CNTs.

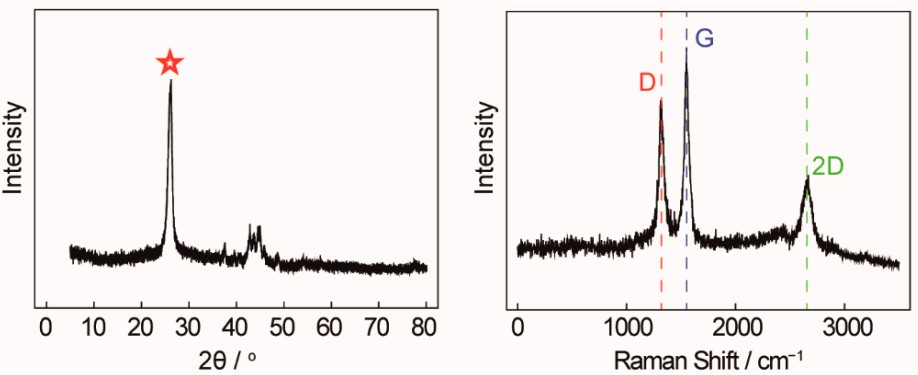

**Figure 4.** (**Left**) XRD of carbon nanotubes. (**Right**) Raman spectra of carbon nanotubes.

### 3.3. The Effects of LiOH Addition on the Cathodic Products

It has been reported that the water electrolysis process powered by renewable energy such as solar and wind energy is a simple process of high purity, with green environmental protection, which is a sustainable method of hydrogen production [35]. In order to further transform $CO_2$ into high-value products, we investigated the effect of water addition on the cathodic products, including gaseous and solid products.

The investigation on the effect of water addition was conducted on the basis of our previous works [22–24,28]. Due to the serious operating difficulty in the input of water or vapor, LiOH is a viable alternative for replacing moisture to supply the hydrogen source. The electrolysis temperature here was 550 °C, which avoids the thermal decomposition of molten salt at high temperatures and the low conductivity caused by too-low temperatures. The electrochemical behavior was studied by cyclic voltammetry to analyze the differences between the pure carbonates and eutectic salts of the carbonates and hydroxides, and the result is shown in Figure 5.

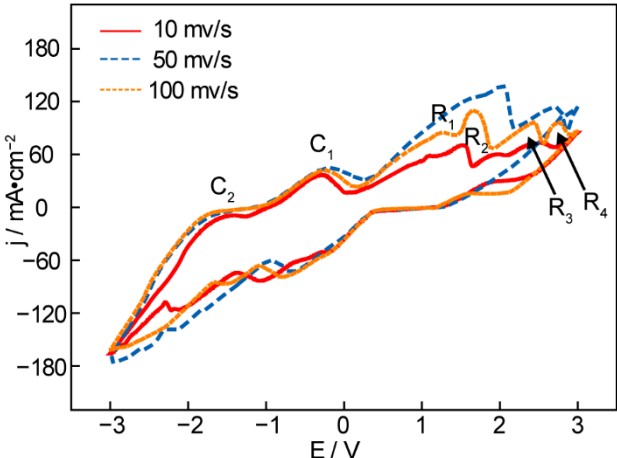

**Figure 5.** Cyclic voltammetry recorded from an Ni anode and Fe cathode in $Li_{1.43}Na_{0.26}K_{0.21}CO_3$ + 0.15LiOH at 550 °C with different scanning rates.

Figure 5 presents the cyclic voltammetry of the molten $Li_2CO_3$-$Na_2CO_3$-$K_2CO_3$ + 0.15LiOH at 550 °C with an Ni anode and Fe cathode at different scanning rates. The scanning result, compared with the preceding CV curve of pure carbonates, is distinguished by a slight reduction current peak, R1, between 0 V and 1 V, suggesting the reduction of $H^+$ to $H_2$. The next large reduction peak, R2, near 1.5 V, can be attributed to the generation of hydrocarbons (the vast majority are methane and a small number are ethane, propane, and other hydrocarbons). According to the previous theoretical calculation, the peak R3 at 2.5 V suggests the generation of CO, and R4 at 2.9 V suggests deposition of the metals K and Na (near 2.7 V, from the theoretical calculation). However, the Li deposition at 550 °C requires a much more significant negative potential and does not appear on the CV. In order to improve the efficiency of electrolysis, the investigation was carried out in a two-electrode cell under an Ar atmosphere, which can protect the electroactivity of two electrodes and expel the gaseous products to the sampling bag through a topside gas-guide tube. $Li_{1.427}Na_{0.359}K_{0.214}CO_3$ and LiOH at the molar ratio of 0.15:1 were mixed and then co-electrolyzed at 550 °C under 2.0 V and 2.5 V for 30 min, and the gaseous products were sampled to test the gas composition by GC.

Below we can see a figure of the gaseous products analysis results, showing that $H_2$, $CH_4$, and CO can be simultaneously detected in a TCD detector after adding LiOH. Furthermore, what catches our attention is the different percentages of $CH_4$ and CO under 1.5 V and 2.5 V. As can be seen in Figure 6, more $CH_4$ was obtained under 1.5 V. However, an unexpected decrease in $CH_4$ and an increase in CO were observed under higher voltages. A higher electrolysis voltage enables the generation of syngas (including $H_2$ and CO), while a lower electrolysis voltage is a proper condition to transform $CO_2/H_2O$ into hydrocarbons (mainly composed of methane). In addition, a small amount of CO was also confirmed by gas chromatography equipped with a TCD detector, as shown in Figure 5. The TCD signal shows well-defined, isolated peaks of $H_2$, $O_2$, $CH_4$, and CO.

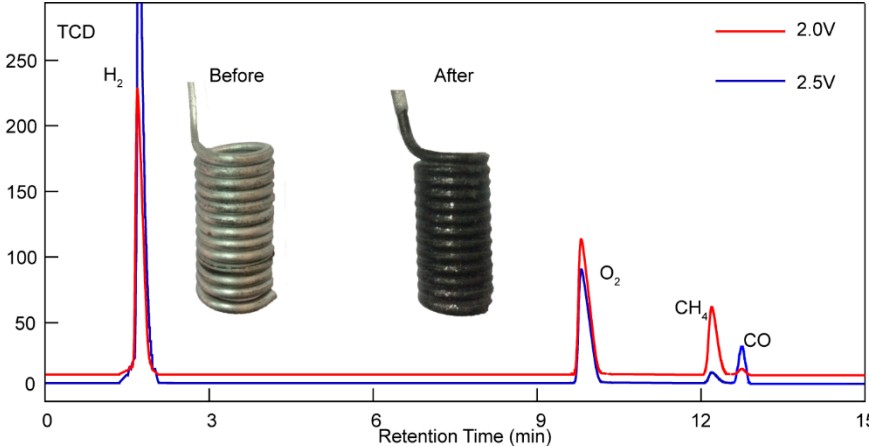

**Figure 6.** Gaseous products analysis results of $Li_{1.427}Na_{0.359}K_{0.214}CO_3$+0.15LiOH under 2.0 V (red line) and 2.5 V (blue line) at 550 °C with a 20 cm$^2$ Ni anode and Fe cathode, as well as a digital photo of the Fe cathode before and after electrolysis.

The core of the co-electrolysis of $CO_2/H_2O$ to produce hydrocarbon-rich fuel gas is the high temperature electrochemical reaction in the molten salt medium. The operating temperature has a significant impact on many parameters of the whole reaction process, such as the molecular activity, theoretical electrolytic potential, molten salt viscosity, molten salt conductivity, gas product composition, and energy conversion efficiency. According to the analysis by theoretical calculation, increasing the operating temperature can effectively reduce the electrolytic potential required by the reaction, but too high a temperature will aggravate the corrosion of the electrode and electrolytic cell and lead to the thermal decomposition of the molten salt. At the same temperature, the electrolytic potential needed to generate $H_2$ is lower than that required to generate $CH_4$. As the data in Table 2

show, at 450 °C, the $H_2$ content in the gas products exceeds 81%, while the $CH_4$ content is only 18.2%. With the increasing temperature, the difference in the electric potential needed to generate $H_2$ and $CH_4$ gradually decreases. The content of hydrocarbons in the product gradually increases with the growing temperature, and the content of $H_2$ gradually decreases. At 500 °C, the peak values of $CH_4$ and $H_2$ are 40.1% and 59.5%, respectively. A high temperature leads to the thermal decomposition of molten salt and excessive side reactions, resulting in a sharp drop in methane gas levels from approximately 40% to less than 27%. In the subsequent experiments, the influence of electrolytic voltage on the composition of the product was mainly studied. The theoretical electrolytic potential required to generate $H_2$ by the electrolysis of hydroxide alone is low, and the $H_2$ content in the gaseous products is close to 100% at an electrolytic voltage of below 1.5 V. The electrolytic voltage gradually increased to meet the electrolytic potential required by $CH_4$, and the content of $CH_4$ in gas products increased significantly. The highest $CH_4$ content was found at 2.0 V, shown as No. 3 in the below table. Continuously improving the voltage applied to the electrode can cause the $CH_4$ content to decrease gradually, and the $H_2$ content will rise again and a certain amount of CO gas will be detected in the gas product. This result is consistent with the theoretical calculation. The highest CO content is up to 33.6% at 550 °C under 3.5 V.

**Table 2.** The results of the gaseous products in $Li_{1.43}Na_{0.26}K_{0.21}CO_3$ + LiOH (90%: 10%, wt%) under different conditions.

| No. | Electrolysis Parameters | | Gas Content | | |
| :-: | :-: | :-: | :-: | :-: | :-: |
| | Temperature (°C) | Voltage (V) | $CH_4$ (%) | CO (%) | $H_2$ (%) |
| 1 | 450 | 2.0 | 18.2 | \ | 81.7 |
| 2 | 500 | 2.0 | 22.1 | \ | 77.6 |
| 3 | 550 | 2.0 | 40.1 | \ | 59.5 |
| 4 | 600 | 2.0 | 30.4 | 0.1 | 69.3 |
| 5 | 650 | 2.0 | 27.3 | 2.4 | 69.9 |
| 6 | 550 | 1.0 | \ | \ | 100 |
| 7 | 550 | 1.5 | \ | \ | 100 |
| 8 | 550 | 2.5 | 13.3 | 9.2 | 77.2 |
| 9 | 550 | 3.0 | 5.4 | 20.2 | 74.0 |
| 10 | 550 | 3.5 | 1.9 | 33.6 | 64.2 |

## 4. Conclusions

It was demonstrated in this paper that molten salt electrochemical technology can convert the greenhouse gas $CO_2$ into high value-added products such as hydrocarbons, hydrogen, oxygen, and carbon materials. In molten carbonates, carbon materials with unique morphologies can be prepared by adjusting reaction parameters or changing electrolyte compositions. CNTs can be synthesized at 750 °C with 10 mA/cm$^2$, initially, and 100 mA/cm$^2$, finally, in a pure $Li_2CO_3$ electrolyte. Further, the SEM data show that the diameters of the CNTs are between 50–80 nm and their purity can reach more than 80%. It was also proven that the Cao and BaO formed during electrolysis are beneficial for synthesizing carbon sphere products with diameters of 200–300 nm at 750 °C with 200 mA/cm$^2$. Honeycomb carbon is obtained in Li-Na-K at 450 °C with a current density of 450 mA/cm$^2$. The addition of hydroxides changes the product from a solid carbon material to a combustible, carbon-based fuel gas. The $CH_4$ content of the product can reach 40.1% at 550 °C under 2.0 V. Meanwhile, by optimizing the electrolytic parameters, the CO content can rise by up to 33.6% at 550 °C under 3.5 V in the electrolyte of $Li_{1.43}Na_{0.26}K_{0.21}CO_3$ + LiOH (90%: 10%, wt%)

**Author Contributions:** Investigation, D.J., Q.J., C.Z. and W.D.; Project administration, H.W. and G.W.; Writing—original draft, D.J.; Writing—review & editing, D.J. All authors have read and agreed to the published version of the manuscript.

**Funding:** This work was supported by the Natural Science Foundation of Heilongjiang Province (LH2021B005) and the Guiding Innovation Foundation of the Northeast Petroleum University (2021YDL-09).

**Institutional Review Board Statement:** Not applicable.

**Informed Consent Statement:** Not applicable.

**Conflicts of Interest:** The authors declare no conflict of interest.

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
