# Peer review of "A New, Efficient Conversion Technology to Transform Ambient CO2 to Valuable, Carbon-Based Fuel via Molten Salt Electrochemistry"

_applsci, doi:10.3390/app12178874_

Round 1
Reviewer 1 Report
The work is very good and of high interest. Unfortunately the grammar usage makes it very hard to read the paper. None of the statements are incorrect however a large amount of work is required to read the paper thoroughly.
Author Response
We thank the reviewer for this valuable suggestion, and the whole manuscript has been polished accordingly.
Reviewer 2 Report
The manuscript entitled “A new efficient conversion technology to transform ambient CO2 to valuable carbon-based fuel via molten salt electrochemistry” describes influence of temperature, voltage and composition of Li-Na-K molten salt electrolytes on the carbon products of electrolysis of CO2.
The paper should be published in Applied Sciences, after minor revision:
1. Throughout the manuscript, the authors often use terms like “high added value” -energy, -products, -carbons etc. Could they please provide any quantification of this added value? Some comparisons or analysis of market costs of various inputs (such as power consumption, raw materials etc) and the produced carbon output will be appreciated.
2. Aside from SEM/EDS, additional characterization (xrd, raman spectroscopy..) of obtained carbons could be of interest to the readers.
3. There are numerous typos and grammatical errors. Lines: 15, 49, 78, 96, 186, 210 etc. The reaction in schematic 1 is not valid, I suggest its removal and using labeled molecule models instead (like in https://doi.org/10.1016/j.ijhydene.2018.09.089). Also, in all CV figures the units of scanning rates are not correct. In figure 5 (GC), the first and most intense peak is not labeled.
Reviewer 3 Report
Reviewer’s Comments:
The manuscript “A new efficient conversion technology to transform ambient CO2 to valuable carbon-based fuel via molten salt electrochemistry” is very interesting work. In this work, alkali metal carbonates (and corresponding hydroxides) are fused and blended to construct a liquid molten salt electrolyte system with excellent performance, which is applied to synthesize carbon materials or carbon-based fuel gas. By regulating the electrolyte composition and electrolysis parameters, Carbon materials with different micro-morphologies and different compositions of carbon-based fuel gas can be pre- pared in a controllable manner. The manuscript is well written, organized, and findings of this manuscript are novel with very good results. The introduction and background are given the premise of the manuscript. The results are consistent with the data and figures presented in the manuscript. While I believe this topic is of great interest to our reader, I think it needs minor revision before it is ready for publishing. So, I recommend this manuscript for publication in this Journal with minor revisions.
1. In abstract, the author should improve the quality of the abstract by adding more results.
2. Keywords: the synthesized system is missing in the keywords. So, modify the keywords.
3. In the introduction part, the author did not elaborate the scientific issues in this field and did not explain well that how their synthesized materials are suitable for these scientific issues.
4. Introduction part is not impressive and systematic. Cite the following articles in the introduction part. (i) 10.1016/j.seppur.2021.119199 (ii) 10.30919/es8d798 (iii) 10.1016/j.electacta.2018.11.088 (iv) 10.1016/j.ceramint.2019.02.123
5. Reaction mechanism…, The author should provide reason about this statement “The movement, reduction and absorption of carbon dioxide are carried out by electrolysis of tetravalent carbon to zero-valent carbon or other carbon-based fuel on the cathode surface”.
6. The authors should explain regarding the recent literature why “The high operating temperature inevitably leads to unwanted thermal decomposition and waste of raw materials”.
7. Material and method: Materials “write all the detail of chemicals in unique format rather than to write individual chemical such as Tin Chloride (SnCl4.5H2O). It should be written as “tin chloride (SnCl4.5H2O, 98%, Sigma)”. Write all the chemicals in this format.
8. In order check the stability of the synthesized materials, the authors should provide the SEM images after CO2 test.
9. Comparison of the present results with other similar findings in the literature should be discussed in more detail. This is necessary in order to place this work together with other work in the field and to give more credibility to the present results.
10. The conclusion part is very week. Improve by adding the results of your studies.
11. The authors should pay more attention to the English grammar, and the abbreviation of journal names in Ref.
Round 2
Reviewer 3 Report
All of the reviewers' concerns were satisfactorily handled by the writers. I am glad to suggest that this article's revision be published as a result.